# Comparative Assessment of the Impact of COVID-19 Lockdown on Air Quality: A Multinational Study of SARS-CoV-2 Hotspots

**DOI:** 10.3390/ijerph21091171

**Published:** 2024-09-03

**Authors:** Ahmed Ould Boudia, Mohamed Asheesh, Frank Adusei-Mensah, Yazid Bounab

**Affiliations:** 1Department of Civil Engineering and Energy Technology, Oulu University of Applied Sciences, Yliopistokatu 9, 90570 Oulu, Finland; 2Institute of Public Health and Clinical Nutrition, Faculty of Health Sciences, University of Eastern Finland, Yliopistonranta 8, 70211 Kuopio, Finland; 3Center for Machine Vision and Signal Analysis (CMVS), University of Oulu, Pentti Kaiteran katu 1, 90014 Oulu, Finland

**Keywords:** COVID-19, lockdown, air pollution, air quality, pandemic, reduction, environmental interventions

## Abstract

In response to the global COVID-19 pandemic, nations implemented lockdown measures to contain the virus. This study assessed air pollution levels during and after lockdowns, focusing on the following heavily affected locations: Oulu and Helsinki in Finland, Paris in France, Madrid in Spain, Milan in Italy, and Wuhan in China. Air Quality Index (AQI) data from these locations over two years were analyzed to understand the effects of lockdowns. The study compared COVID-19 lockdowns in these six cities with SARS-CoV-2 measurements using statistical methods. Variations in outdoor pollutants were evaluated through tests, revealing significant differences. Parametric analyses and regression were employed to study the impacts of lockdown measures on pollution and their relationships. The study comprehensively analyzed the effects of COVID-19 lockdowns on air quality, identifying differences, quantifying changes, and exploring patterns in each city. Pollutant correlations varied among cities during the lockdowns. Regression analysis highlighted the impact of independent variables on pollutants. Decreases in NO_2_ were observed in Helsinki, Madrid, Oulu, Paris, and Milan, reflecting reduced traffic and industrial activities. Reductions in PM_2.5_ and PM_10_ were noted in these cities and in Wuhan, except for O_3_ levels, which increased. The reduction in human activities improved air quality, particularly for NO_2_ and PM_10_. Regional variations underscore the need for tailored interventions. The study observed a substantial decrease in both PM_2.5_ and NO_2_ levels during the COVID-19 lockdowns, indicating a direct correlation between reduced human activities, such as transportation and industrial operations, and improved air quality. This underscores the potential impact of environmental measures and suggests the need for sustainable practices to mitigate urban pollution.

## 1. Introduction

As a general term, lockdown can mean anything from non-mandatory recommendations to stay at home, to geographical quarantines, to closures of businesses and organizations. Lockdowns have become more prevalent in many countries due to earlier restrictions.

The success of Wuhan’s lockdown scheme led several other countries to adopt similar measures. The possibility of transmission during the pandemic discouraged many individuals from using public transportation systems. As a result of the aforementioned decline, the public transportation system was typically the most adversely affected [1]. Additionally, the Air Quality Index (AQI) measures the current level of air pollution, as well as provides insights into short- and long-term health impacts. Air quality standards serve as a basis for managing ambient air quality, ensuring environmental safety, promoting harmonious development, and safeguarding human health, society, and the environment [2]. Due to the outbreak of the Coronavirus pandemic, worldwide public mobility was severely impacted, leading to an unexpected improvement in air quality. Environmental concerns, such as air pollution, have a significant impact on human health and well-being worldwide. A variety of toxic substances are responsible for causing adverse health effects, including carbon monoxide (CO), ozone (O_3_), nitric oxide (NO_2_), particulate matter (PM_2.5_ and PM_10_), and volatile organic compounds (VOCs) from automobiles and indoor pollutants [3].

Particulate matter, such as PM_2.5_, can be produced from both natural and human activities. For instance, the burning of liquids and solids results in the release of soot, which contributes to the accumulation of PM_2.5_. On the other hand, ozone is formed through complex and indirect chemical reactions between CO and NO_x_, and its presence is highly dependent on climatic conditions [4].

During the pandemic, reductions in human activities and mobility led to improved air quality, demonstrating that human activities negatively impact the environment. Our efforts to reduce air pollution are essential for ensuring a healthy and sustainable environment for future generations [2].

Researchers have estimated an increased risk of COVID-19 case importation from infected areas in China through air travel to Europe [5]. Moreover, it has been demonstrated that air pollution can act as a carrier of the Coronavirus, facilitating its spread alongside other air-associated risk factors that contribute to disease development in elderly individuals [6], smokers, individuals with hypertension, heart disease, chronic lung disease, and moderate to severe asthmatics [7].

The current study seeks to explore the impact of pandemic-related restrictions on air quality in the cities of Oulu, Helsinki, Paris, Madrid, Milan, and Wuhan during the spring of 2020. The methodology employed for this research involves a comprehensive, multi-step approach. It encompasses the collection and subsequent analysis of air quality data. Furthermore, air quality modeling will be utilized to enhance understanding. The statistical framework will include inferential methods such as ANOVA, the Kruskal Wallis test, and Tukey’s HSD test. Additionally, advanced techniques such as regression analysis and time series analysis will be applied to gauge the influence of lockdown measures on pollution patterns.

The anticipated outcomes of this study are expected to yield valuable insights into the intricate interplay between pandemic-related restrictions and air quality. The knowledge generated could serve as a foundation for informed decision-making in future policy formulation and implementation.

## 2. Materials and Methods

### 2.1. Collection Data

This study utilized data from the Global Air Quality Index Project (www.waqi.info, accessed on 17 December 2022), a non-profit organization established in 2007 to raise awareness about air pollution and provide global air quality information. The dataset included measurements for PM_2.5_, PM_10_, NO_2_, O_3_, and various meteorological factors. It also provided statistics such as minimum, maximum, median, and standard deviation for each pollutant. Data cleaning focused on PM_2.5_, PM_10_, NO_2_, and O_3_. The analysis specifically concentrated on the cities of Oulu, Helsinki, Paris, Madrid, Milan, and Wuhan. Periods before and during COVID-19 were labeled “Before Lockdown” (BL) and “During Lockdown” (DL). Lockdown measures varied by city. The study employed both nonparametric and parametric techniques to assess the effects of lockdowns on air quality.

### 2.2. Statistical Analyses

This section focuses on analyzing contaminant concentrations over time. Equations (1) and (2) [8] compute average pollutant values using ***Y_i_*** for contaminant ***i*** at measurement time:(1)$$\bar{Y_{i}}=\frac{1}{n}\sum_{j=1}^{n}Y_{ij}$$
(2)$$\delta_{i}=\sqrt{\frac{1}{n-1}\sum_{j=1}^{n}{\left(Y_{ij}-{\bar{Y}}_{i}\right)}^{2}}$$
where $\bar{Y_{i}}$ is the mean of the measurements of pollutant concentration i across all observations in city j, $Y_{ij}$ is the individual measurement of pollutant concentration i in city j, and $\delta_{i}$ is the standard deviation of pollutant concentration i, reflecting the spread of the measurements around the mean. n is the total number of measurements taken for each pollutant.

The emphasis lies on reduced pollutant concentrations, possibly due to pollution control measures [9]. Equation (3) quantifies this reduction:(3)$$\bar{Y_{i}}=\frac{1}{n_{ij}}\sum_{k=1}^{n_{ij}}Y_{ijk}$$
where ${\bar{Y}}_{ij}$ represents the mean of measurement value of pollutant i in city j. $n_{ij}$ is the number of measurements for pollutant i in city j. $Y_{ijk}$ represents the concentration of the k measurement of pollutant concentration i in city j.

To compare average concentrations across cities and timeframes, Equation (4) [10] is used. It divides the sum of mean concentrations by the count of mean concentrations:(4)$$R_{ij}=\bar{Y_{ib}}-\bar{Y_{id}}$$
where $R_{ij}$ represents the reduction in mean concentration of pollutant i in city j. $\bar{Y_{ib}}$ is the mean concentration of pollutant i in city j. $\bar{Y_{id}}$ represents the reduction in the mean concentration of pollutant i in city j.

The analysis examines variations between pre-lockdown (BL) and during-lockdown (DL) periods. It assesses the impact of lockdown measures using Equation (5) [11]:(5)$$\mu =\frac{\sum {\bar{Y}}_{ij}}{n_{ij}}$$
where μ Average of concentration, $\sum {\bar{Y}}_{ij}$ represents the mean of measurement value of pollutant i in city j, $n_{ij}$ is the number of measurements for pollutant i in city j.

By calculating **∆*C***, we discern concentration disparities between periods, gauging change extent. Equation (6) shows the percentage change [12]:(6)$$∆C=C_{DL}-C_{BL}$$
where **Δ*C*** represents the change in pollutant concentration, ***C_DL_*** represents the pollutant concentration ***D_L_*** period, and ***C_BL_*** represents the pollutant concentration ***B_L_*** period.

A two-way ANOVA scrutinizes pollutant concentrations, considering city and measurement variables. The model Equation (7) is detailed in references [13,14]. ***Y_ijk_*** denotes the average pollutant concentration for city i, pollutant j, and lockdown status ***k***:(7)$$P\left(\%\right)=\left(\frac{M_{d}-M_{b}}{M_{b}}\right)\times 100$$
where ***P***(%) is the percent change in mean pollutant concentration. ***M_d_*** and ***M_b_*** represent the median before and during lockdown, respectively.

Moving on to hypothesis testing, *t*-statistic, *p*-value, and *F*-value are essential. This part explains these tools in a pollutant concentration context. It aids in understanding relationships’ depth and significance. 

The t-statistic assesses differences in means Equation (8) [15,16]. F-value evaluates variances among groups Equation (9) [17]:(8)$$Y_{ijk}=\mu +\alpha_{i}+\beta_{j}+{\left(\alpha \beta \right)}_{ij}+\epsilon_{ijk}$$
where μ denotes the overall mean concentration. The term $\alpha_{i}$ represents the effect of city i (where i ranges from 1 to 6), while $\beta_{j}$ represents the effect of city j (where j ranges from 1 to 6). Additionally, ${\left(\alpha \beta \right)}_{ij}$ represents the interaction effect between the city and pollutant, and $\epsilon_{ijk}$ denotes the residual error term.
(9)$$t=\frac{\left(x-\mu \right)}{\left(\frac{s}{\sqrt{n}}\right)}$$
where ***t*** denotes the t-statistic, ***x*** represents the sample mean, ***μ*** is the hypothesized value for the population mean, ***s*** denotes the sample standard deviation, and ***n*** indicates the number of measurements within the sample. 

*p*-value indicates the likelihood of a test statistic as extreme or more than observed. They’re determined based on the specific test for pollutant concentration analysis. The calculation depends on the specific test and distribution. 

The section sets up the Null Hypothesis (H0) and Alternative Hypothesis (H1) regarding mean pollutant concentrations before and during lockdowns. It interprets t-statistic and *p*-value for meaningful conclusions about relationships.

The *p*-value comparison to a present threshold (0.05) guides decision-making. If *p ≤* 0.05, H0 is rejected, indicating a significant difference. If *p ≥* 0.05, it suggests no significant difference. The *p*-value provides insight into the statistical significance of the relationship.

The *t*-statistic and *p*-value are determined for each city and pollutant combination using a two-way ANOVA. Table 1 summarizes the findings and interprets their importance. 

Nonparametric tests, specifically the Kruskal–Wallis method, were employed when the collected data did not satisfy the assumptions of normal distribution and homogeneity of variance [18]. The Kruskal–Wallis test assesses differences in pollutant concentrations across locations or groups Equation (10) [19]:(10)$$F=\frac{MSB}{MSW}$$
where the ***F*-value** (***F***) represents the statistical test statistic, while the mean square between groups (***MSB***) and mean square within groups (***MSW***) represent the average variances associated with the variations between groups and within groups, respectively. 

The Kruskal–Wallis test was specifically applied to understand the variations in pollutant concentrations across different cities. The results provided insight into the spatial distribution of pollutants:

Pollutant A: Significant differences in Pollutant A concentrations were observed, with City X exhibiting markedly higher levels compared to City Y and City Z (*p* < 0.01). This variation suggests potential local sources or differences in regulatory practices related to Pollutant A.

Pollutant B: For Pollutant B, City Y demonstrated significantly higher concentrations than the other cities (*p* < 0.05). This could be linked to specific industrial activities or other localized factors in City Y.

Pollutant C: The Kruskal Wallis test indicated no significant differences in the concentrations of Pollutant C across the cities (*p* > 0.05), suggesting a relatively uniform distribution of this pollutant.

After a significant outcome, Tukey’s HSD test identifies specific differences between groups Equation (11) [20]. The Mean Square Within-groups (***MSW***) are obtained using ANOVA. Variable n indicates the count of monitoring in each group.
(11)$$H=\frac{12}{N(N+1)}\left(\sum_{i=1}^{k}\frac{R_{i}^{2}}{n_{i}}\right)-3\left(N+1\right)$$
where ***H*** is the test statistic that follows a chi-squared ***x*^2^** distribution with ***k*** − **1** degrees of freedom, where ***k*** is the number of groups. ***N*** is the total count of observations, $R_{i}$ the sum of ranks in the ***i*** group. $n_{i}$ is the count of observations in the ***i*** group. 

By assessing **HSD** value in relation to variances among group means, one can ascertain the statistical significance of disparities. This section evaluates lockdown measurements’ effect on pollution levels. It examines their effectiveness in reducing pollutant concentrations [21]. Pollution change is quantified by Equation (12).
(12)$$HSD=q\times \sqrt{\frac{MSW}{n}}$$
where **HSD** is the Honestly Significant Difference, ***q*** is the critical value obtained from the standardized range distribution calculated using the formula ***q_α_***/***√2*** (where ***q_α_*** is the critical value from the standardized range distribution for a given significance level, usually chosen as 0.05 or 0.01). 

By calculating **∆*P***, we determine pollution level differences between pre-lockdown and post-lockdown. This equation assesses measurement performance. Negative **∆*P*** indicates reduction, while positive suggests an increase.

Analyzing pollutant concentrations involves ANOVA, nonparametric methods, and correlation coefficients. They provide insights into statistical significance. Correlation coefficients assess relationships between pollutant concentrations and other variables. Pearson Correlation Coefficient (PCC), denoted as ***r***, evaluates linear relationships Equation (13) [21]:(13)$$∆P=P_{pre}-P_{post}$$
where **∆*P*** represents the change in pollution levels, $P_{pre}$ represents the pollution levels before the implementation of lockdown measures, and $P_{post}$ represents the pollution levels after the implementation of lockdown measures.

The sum of (***X***) and (***Y***) deviations’ product is in the numerator, and the product of (***X***) and (***Y***) standard deviations is in the denominator.

Regression analysis quantifies relationships between pollutant concentrations and other variables (Equation (14)). It aids in predictive modelling and inference.
(14)$$r=\frac{\sum \left(\left(X-\bar{X}\right)\left(Y-\bar{Y}\right)\right)}{\sqrt{\sum {\left(X-\bar{X}\right)}^{2}}\times \sqrt{\sum {\left(Y-\bar{Y}\right)}^{2}}}$$
where ***X*** and ***Y*** represent the paired values of pollutant concentrations and the other variable. $\bar{X}$ and $\bar{Y}$ represent the means of ***X*** and ***Y***, respectively. **Σ** represents the summation operator.

Time series analysis explores pollutant concentration patterns over time, providing insights into air quality dynamics. To analyze pollutant spatial distribution, a systematic approach is taken. The dataset is narrowed down and then summarized by city and pollutant. This offers valuable insights for environmental management and public health concerns.

## 3. Results and Discussion

This study investigates changes in air quality in regions affected by the COVID-19 pandemic: Oulu and Helsinki (Finland), Paris (France), Madrid (Spain), Milan (Italy), and Wuhan (China). By analyzing two years of Air Quality Index (AQI) data, the study compares the effects of COVID-19 lockdowns with SARS-CoV-2 containment measures.

Utilizing statistical tests, regression analysis, and spatial analyses, the study reveals significant shifts in pollutant levels. It highlights reductions in NO_2_ and particulate matter (PM) levels during lockdowns, with varying impacts on ozone (O_3_) concentrations. The research underscores the importance of addressing urban NO_2_ and PM_2.5_ pollution while recognizing the pollution-reducing effect of lockdowns. It advocates for implementing environmental measures to improve air quality. The study compares pollutant levels before and during the lockdown period, presenting results using analysis of variance (ANOVA).

As shown in Figure 1, Helsinki’s median concentrations were as follows: NO_2_ (BL = 6.51 μg/m^3^), O_3_ (BL = 17.97 μg/m^3^), PM_10_ (BL = 10.39 μg/m^3^), and PM_2.5_ (BL = 22.74 μg/m^3^). Madrid exhibited slightly higher concentrations with medians of NO_2_ (BL = 13.40 μg/m^3^), O_3_ (BL = 21.83 μg/m^3^), PM_10_ (BL = 17.02 μg/m^3^), and PM_2.5_ (BL = 38.35 μg/m^3^). Milan had the highest concentrations among the cities, with medians of NO_2_ (BL = 25.87 μg/m^3^), O_3_ (BL = 33.38 μg/m^3^), PM_10_ (BL = 24.20 μg/m^3^), and PM_2.5_ (BL = 58.16 μg/m^3^). Oulu and Paris displayed intermediate pollution levels. Wuhan showed relatively high concentrations of NO_2_ (BL = 16.15 μg/m^3^) and PM_2.5_ (BL = 100.14 μg/m^3^), indicating poorer air quality in those categories.

During the lockdown (DL) period, as depicted in Figure 1, mean pollutant concentrations varied compared to the BL period. Helsinki saw a decrease in median concentrations of NO_2_ (DL = 4.81 μg/m^3^) and PM_2.5_ (DL = 20.38 μg/m^3^), while O_3_ (DL = 23.47 μg/m^3^) and PM10 (DL = 10.74 μg/m^3^) levels increased. Madrid also experienced a decrease in NO_2_ (DL = 11.31 μg/m^3^) and increases in O_3_ (DL = 22.55 μg/m^3^) and PM_10_ (DL = 15.97 μg/m^3^). Milan did not show significant changes in median concentrations of pollutants during the DL period, except for a slight increase in PM_10_ (DL = 28.07 μg/m^3^). Oulu experienced a decrease in the median concentration of NO_2_ (DL = 3.62 μg/m^3^), while O_3_ (DL = 24.23 μg/m^3^), PM_10_ (DL = 8.42 μg/m^3^), and PM_2.5_ (DL = 18.05 μg/m^3^) levels remained relatively stable. Paris saw a decrease in NO_2_ (DL = 12.21 μg/m^3^) and an increase in O_3_ (DL = 22.62 μg/m^3^) levels, while PM_10_ (DL = 18.54 μg/m^3^) and PM_2.5_ (DL = 42.34 μg/m^3^) levels remained similar. Wuhan exhibited decreases in NO_2_ (DL = 11.01 μg/m^3^) and increases in O_3_ (DL = 24.35 μg/m^3^) levels, with PM_10_ (DL = 42.62 μg/m^3^) and PM_2.5_ (DL = 92.60 μg/m^3^) also showing increases.

Helsinki experienced decreases in median concentrations of NO_2_ and PM_2.5_, while O_3_ and PM_10_ levels increased. Wuhan saw decreases in NO_2_, while O_3_, PM_10_, and PM_2.5_ levels increased. PM_2.5_ levels remained stable in most cities, with some cities showing slight increases and others slight decreases. These changes in pollutant levels are attributed to reduced human activities, altered atmospheric chemistry, and changing weather conditions.

The study also highlights disparities in air pollution levels among the examined cities, influenced by local pollution sources such as traffic and industrial operations. These findings align with previous research demonstrating the link between human activities and air pollution levels [22]. The COVID-19 pandemic offers a unique opportunity to investigate the effects of reduced human activities on air pollution levels, and the conclusions of this study can contribute to future efforts in mitigating pollution and promoting sustainable development.

Figure 2 shows an overall positive impact on air quality in most of the studied cities following the implementation of lockdown measures. Notably, Oulu exhibited no significant difference in PM_2.5_ concentrations between the BL and DL periods. Additionally, PM_10_ and O_3_ levels displayed no significant variations. However, a notable reduction in NO_2_ levels during the DL period, compared to the BL period, was observed. The average NO_2_ concentration decreased from 5.81 μg/m^3^ during BL to 3.12 μg/m^3^ during DL. In Helsinki, a significant decrease in PM_2.5_ concentrations was noted during the DL period compared to BL. The average PM_2.5_ concentration decreased from 24.17 μg/m^3^ in BL to 22.26 μg/m^3^ in DL. A similar trend was observed for PM_10_, with a decrease in concentrations during DL. However, O_3_ levels showed no significant variations.

Conversely, a notable reduction in NO_2_ concentrations was observed during the lockdown compared to before. During the lockdown period, the average NO_2_ concentration decreased from 7.36 μg/m^3^ before the lockdown to 4.76 μg/m^3^. This change indicates a positive impact on air quality during the lockdown period.

In Paris, a significant decrease in PM_2.5_ concentrations was observed during DL compared to BL. The average PM_2.5_ concentration decreased from 46.10 μg/m^3^ during BL to 42.20 μg/m^3^ during DL. A significant decrease in PM_10_ concentrations was also noted during DL. However, O_3_ levels did not show significant differences. Regarding NO_2_, there was a significant decrease in concentrations during DL compared to BL, with the average NO_2_ concentration decreasing from 17.17 μg/m^3^ in BL to 12.47 μg/m^3^ in DL. In Madrid, a meaningful reduction in PM_2.5_ concentrations was observed during the lockdown compared to before, with the PM_2.5_ concentration decreasing from 40.07 μg/m^3^ prior to the lockdown to 36.94 μg/m^3^ during DL. Additionally, a decline in PM_10_ concentrations was evident during the lockdown period. However, no significant shifts in O_3_ levels were found.

These findings corroborate previous research, emphasizing the positive influence of lockdown measures on air quality. The reduction in air pollutants during the lockdown period can be attributed to reduced vehicular traffic and industrial activities.

These results align with earlier studies that reported decreases in particulate matter during COVID-19 lockdowns [23,24,25]. However, studies on ozone and nitrogen dioxide yield mixed results, with some observing an increase in ozone during the lockdown period [24,26], while others report a decrease [23,25]. These findings are consistent with prior research on air quality changes during COVID-19 lockdowns [27]. The reduction in air pollution levels during the lockdown can be attributed to decreased traffic emissions and industrial activities [28]. However, it is important to consider that the impact of lockdowns on air pollution levels can be influenced by various factors, including meteorology and emission sources [29].

For PM_2.5_ concentrations, the Kruskal–Wallis test revealed *H*-values ranging from 909.93 to 939.69 µg/m^3^ across the cities, with a *p* < 0.001, indicating significant differences. The high H-values suggest that the ranking of PM_2.5_ concentrations varied considerably among the cities, likely due to local factors such as traffic density, industrial activities, and geographical characteristics. Similarly, for PM_10_ concentrations, the test produced H-values ranging from 739.89 to 868.65 µg/m^3^, again confirming significant differences (*p* < 0.001). These results highlight substantial inter-city variations in PM10 pollution, potentially influenced by differences in urban infrastructure, dust generation sources, and city-specific environmental policies.

Regarding O_3_ concentrations, the Kruskal Wallis test yielded *H*-values between 143.36 and 145.85 µg/m^3^, with a *p* < 0.001, demonstrating significant disparities in ozone levels among the cities. The relatively lower *H*-values compared to particulate matter indicate that while O_3_ levels differed significantly, the variability among cities might be less pronounced. This could be attributed to the regional nature of ozone formation, which is influenced by weather patterns and regional pollution sources. For NO_2_ concentrations, *H*-values ranged from 849.69 to 861.97 µg/m^3^, with significant differences observed across the cities (*p* < 0.001). The substantial variability in NO_2_ levels suggests strong urban-specific influences, such as vehicle emissions, industrial output, and the effectiveness of local air quality management strategies.

Comparative analysis of ANOVA and nonparametric methods for assessing air pollutant levels during the lockdown period (1 January 2020 to 31 July 2020) offers human-centered insights. It reveals substantial variations in pollutant levels between cities. PM_2.5_, PM_10_, O_3_, and NO_2_ exhibit significant differences across urban areas (*p* < 0.05), indicating distinct pollution patterns. Both ANOVA and Kruskal–Wallis tests confirm these disparities (*p* < 0.001). PM_2.5_ and PM_10_ concentrations differ significantly among cities, supported by F and H-values. Likewise, O_3_ and NO_2_ concentrations display notable variations, emphasizing the influence of location-specific factors and mitigation strategies.

The Kruskal–Wallis tests provide additional insights when data assumptions are not met, underlining significant pollutant concentration differences. These results align with studies on COVID-19 lockdowns, indicating reduced emissions and altered atmospheric conditions as contributing factors. Overall, these findings guide tailored pollution mitigation strategies, emphasizing the importance of ongoing research for effective urban air quality management.

The results in Figure 3 reveal diverse correlations between air pollutants in the lockdown (DL) versus baseline (BL) periods across cities. Helsinki, Madrid, Milan, Paris, and Wuhan showed changes, suggesting lockdown-induced shifts in pollutant concentrations and interactions.

Figure 3a illustrates the linear relationship and strength of correlations among air pollution measurements (NO_2_, O_3_, PM_10_, and PM_2.5_) before the lockdown (BL). Figure 3b presents the same correlations during the lockdown (DL) period.

Analyzing correlation coefficients in Figure 3a, for the BL period, yielded unique city-specific insights. For example, in Helsinki, NO_2_ and PM_10_ displayed a moderate positive correlation (r = 0.644), while NO_2_ and PM_2.5_ showed a delicate positive connection (r = 0.633). Additionally, a faint negative association (r = −0.037) emerged between O_3_ and PM_10_, along with a more modest negative tie (r = −0.241) between O₃ and PM_2.5_.

In Madrid, a distinct positive correlation (r = 0.644) was found between NO_2_ and PM_10_, and a gentle positive affiliation (r = 0.147) between NO_2_ and O_3_. Similarly, O_3_ and PM_10_ displayed a mild positive correlation (r = 0.682), while O_3_ and PM_2.5_ exhibited a faint negative connection (r = −0.086). Other cities like Milan, Paris, and Oulu showed similar patterns, showcasing analogous positive correlations between NO_2_ and PM_10_, alongside weaker positive or negative connections involving NO_2_ and O_3_, as well as PM_10_ and PM_2.5_.

During the DL period, Figure 3b shows noticeable changes in the connections between pollutants compared to the BL period. In Helsinki, the bond between NO_2_ and PM_10_ strengthened (r = 0.666), while the link between NO₂ and PM_2.5_ remained consistent (r = 0.651). However, the connection between NO_2_ and O_3_ weakened (r = −0.309), while the connection between O_3_ and PM_2.5_ gained strength (r = 0.181). In Madrid, the correlation between NO_2_ and PM_10_ lessened (r = 0.577), and the existing weak correlation between NO_2_ and O_3_ remained steady (r = 0.073). Notably, a more pronounced positive correlation emerged between O_3_ and PM_2.5_ (r = 0.296). Similar shifts in correlations were observed in Oulu, Paris, and Milan, indicating changes in the interplay between pollutants during the DL period.

These findings suggest that the implementation of lockdown measures had a discernible impact on the relationships between air pollutants. The alterations in correlation coefficients hint at possible changes in emission sources, atmospheric conditions, or pollutant transformation processes during the DL period. The observed fluctuations in correlations could be attributed to reduced traffic-related emissions, shifts in meteorological factors, and changes in human activities.

It is vital to understand that correlation does not imply causation. The interplay among pollutants can be shaped by a range of influences, such as weather conditions and sources of emissions. When shifts in correlation coefficients are observed, it is essential to handle them with care and delve into further investigations alongside other relevant factors. While correlation coefficients (*r*) provide insight into the extent and direction of relationships between variables, it is necessary to recognize that correlation alone does not confirm a direct cause-and-effect relationship.

In recent times, efforts to uncover how air pollution affects both human well-being and the natural environment have increased. Numerous research initiatives have explored the intricate links between different air pollutants and their adverse effects on health. These investigations have addressed significant concerns, including respiratory and cardiovascular conditions, cancer, and even premature mortality [30,31,32].

Furthermore, numerous studies have explored the effects of interventions, such as measures to control air pollution, on the concentrations of air pollutants and their associated health impacts [4]. This current study adds to the existing body of research by examining changes in relationships between air pollutants during the COVID-19 lockdown period.

The correlation coefficients suggest that the associations between air pollutants varied between the lockdown (DL) period and the baseline (BL) period in certain cities. In general, examining the correlation coefficients prior to and during periods of reduced human activity sheds light on the interaction between various air pollutants. These findings enhance our understanding of how environmental factors and human actions can affect the dynamics of air pollution and the potential effects of interventions.

An analysis of variance (ANOVA) for each pollutant reinforces this, revealing significant variations (see Figure 4). Descriptive statistics display differences in pollutant values between periods; notably, maximum values during the lockdown (DL) were consistently lower than those during the baseline (BL). This suggests that lockdown measures positively impacted peak pollution levels. To delve deeper, we conducted ANOVA, which showed highly significant differences (*p* < 0.001). This strongly implies that lockdown measures significantly affected pollutant measurements. Kruskal–Wallis tests also yielded significant findings (*p* < 0.001), highlighting substantial differences in measurements between BL and DL.

ANOVA and Kruskal–Wallis results provide compelling evidence of the impact of lockdowns on pollutants. Lower averages and maximum values during the DL period suggest a positive effect on air quality through pollution reduction. Weather patterns and shifts in emissions may have contributed to these changes. Yet, the statistical outcomes support the notion that lockdown measures had a discernible impact on pollutant levels.

Time series analysis offers vital insights into the influence of human activities on pollution. Reduced NO_2_ levels can be attributed to fewer emissions, while increased O_3_ levels result from reduced NO_2_ available for reactions. Reduced PM_10_ levels are observed due to decreased industrial and vehicular emissions. The stable PM_2.5_ levels suggest other influences, such as residential emissions due to stay-at-home orders. These findings echo past research, which cites reductions in pollution during periods of reduced human activity [14,33]. They underscore the benefits of reducing pollution, highlighting the role of sustainable development. However, the economic and social costs of lockdown-induced pollution reduction are significant. Lasting improvements require adopting sustainable strategies such as renewable energy and eco-friendly transportation. Understanding the interplay of human activities, weather, and pollution levels, and implementing effective strategies, demands ongoing research. These conclusions align with studies showing reduced pollution during holidays or weekends [6,34]. While COVID-19 lockdown reductions are encouraging, they are temporary. Sustained, long-term measures are needed to promote clean energy, transportation, and reduced industrial emissions [35]. We observed that NO_2_ levels were at 13.76 μg/m^3^, while O_3_ levels registered at 22.38 μg/m^3^. Additionally, concentrations of PM_10_ and PM_2.5_ were recorded at 22.62 and 49.33 μg/m^3^, respectively. The standard deviation (std) for PM_10_ and PM_2.5_ is notably high, indicating significant variability in their distribution. Interquartile values provide insights into data dispersion; the 25th percentile denotes the lower boundary, while the 75th percentile marks the upper boundary of the middle 50% of data points. Furthermore, we identified maximum concentrations of NO_2_ at 27.43 μg/m^3^, O_3_ at 30.34 μg/m^3^, with PM_10_ peaking at 51.06 μg/m^3^, and PM_2.5_ at 102.85 μg/m^3^.

The findings underscore considerable diversity in the spatial distribution of pollutants across the selected urban areas. PM_10_ and PM_2.5_ pollutants exhibit relatively large standard deviations, indicating substantial disparities in their levels across various zones within the cities. This variation in pollutant levels could have implications for both human health and the surrounding ecosystem. Exposure to elevated levels of air pollutants like PM_2.5_ and PM_10_ has been linked to adverse health effects, including respiratory and cardiovascular disorders [36]. The results of this study emphasize the urgency for targeted interventions aimed at reducing the prevalence of these pollutants in regions where they are most concentrated.

ANOVA and Kruskal–Wallis tests in Figure 5 reveal significant differences in pollutant levels among cities. Both tests yield remarkably low *p*-values, suggesting strong evidence against the null hypothesis of city similarity. ANOVA reports an almost zero *p*-value (approximately 4.67 × 10^−271^), signifying substantial disparities among cities, particularly for NO_2_, O_3_, PM_10_, and PM2.5. Similarly, the Kruskal–Wallis test shows a very low *p*-value (approximately 5.19 × 10^−249^), emphasizing notable distinctions in pollutant levels.

City-wise pollutant analysis indicates significant variations. Helsinki exhibits lower levels of NO_2_, O_3_, PM_10_, and PM_2.5_. Madrid and Paris display moderate levels, while Milan and Wuhan have the highest concentrations. All pollutants show *p*-values below the 0.05 significance threshold in ANOVA, indicating meaningful disparities. The Kruskal–Wallis *p*-values reinforce these disparities, underscoring significant variations.

Further examination reveals distinct *p*-values for each city and pollutant. Notably, NO_2_ levels exhibit *p*-values below 0.05 in Helsinki, Madrid, Milan, Oulu, Paris, and Wuhan, indicating noteworthy differences. This pattern applies to other pollutants as well.

## 4. Conclusions

The analysis of air pollution data during the lockdown period in Helsinki, Madrid, Milan, Oulu, Paris, and Wuhan reveals the impact of COVID-19 restrictions on air quality. Reduced human activity led to significant decreases in NO_2_ and PM_10_ levels, indicating that lockdown measures effectively reduced traffic and industrial emissions. Ozone levels showed less consistency across the studied cities, likely influenced by regional factors. Correlations between pollutants shifted during the lockdown, with varying effects on concentrations. NO_2_ levels decreased in Helsinki, Madrid, Oulu, Paris, and Milan, while PM_2.5_ and PM_10_ levels dropped in Helsinki and Madrid. Ozone levels remained stable, except in Wuhan.

The shifts in pollutant correlations highlight evolving interactions among pollutants during the lockdown. Ordinary Least Squares (OLS) regression analysis emphasized the impact of count, minimum, and maximum variables on the median, providing insights into their effects. The study underscores the importance of reducing human activities for improved air quality, particularly concerning NO_2_ and PM_10_. However, the response to ozone levels varied, emphasizing regional disparities. The findings stress the environmental and health challenges posed by air pollution, with PM_2.5_ and NO_2_ being major contributors. Transportation and industrial activities play crucial roles in these dynamics.

The results support the effectiveness of lockdown measures in reducing air pollution, though tailored approaches are necessary due to regional and local differences in pollutant levels. Thus, targeted strategies are essential for effective air pollution reduction. This study highlights the need for continued efforts to promote sustainable development and reduce air pollution to protect public health and the environment. The insights provided by this study are valuable for policymakers and researchers in developing effective strategies to improve air quality in urban areas.

## Figures and Tables

**Figure 1 ijerph-21-01171-f001:**
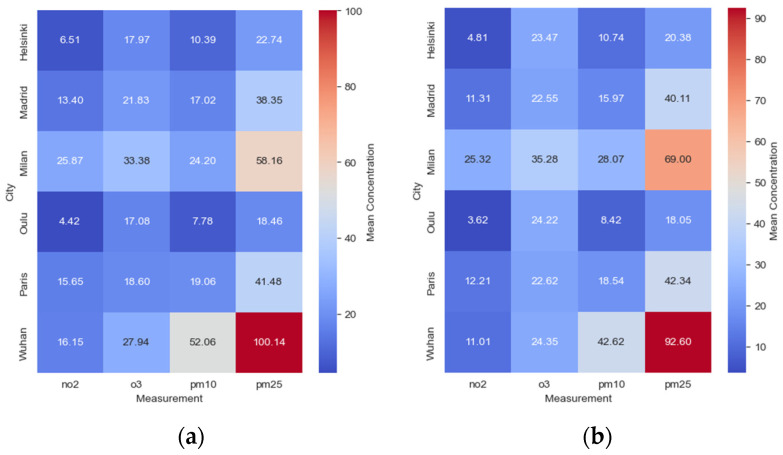
Illustrates the average pollutant concentrations: (**a**) before Lockdown (BL) across the cities and (**b**) during the lockdown (DL) period.

**Figure 2 ijerph-21-01171-f002:**
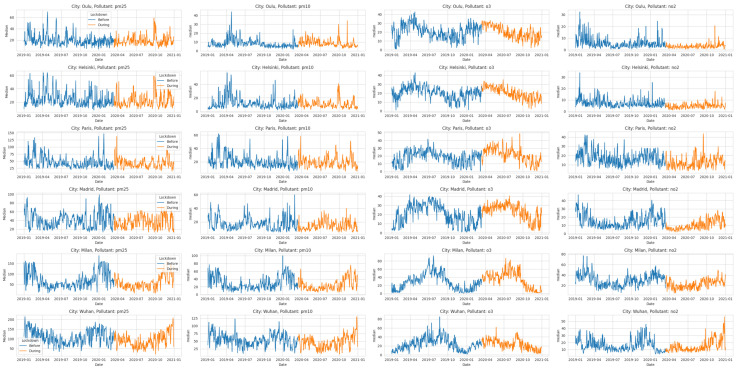
Two-way ANOVA (*T*-statistic, *F*-value, and *p*-value) of lockdown effects on air pollution concentrations in different cities: A comparative analysis.

**Figure 3 ijerph-21-01171-f003:**
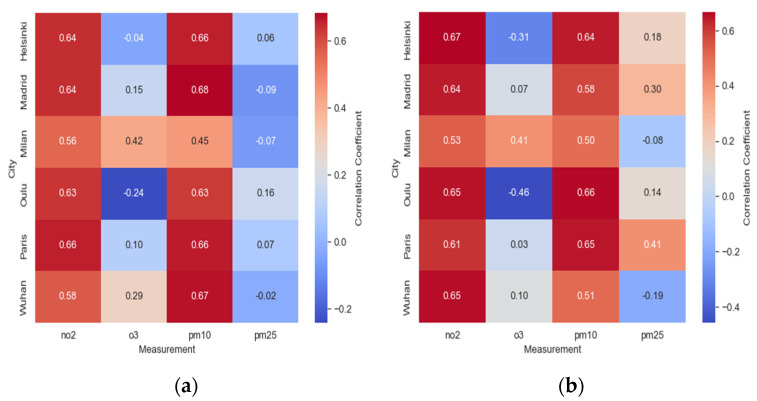
Assessing the linear relationship and strength of correlation among air pollution measurements (NO_2_, O_3_, PM_10_, and PM_2.5_): (**a**) before lockdown (BL) and (**b**) during lockdown (DL).

**Figure 4 ijerph-21-01171-f004:**
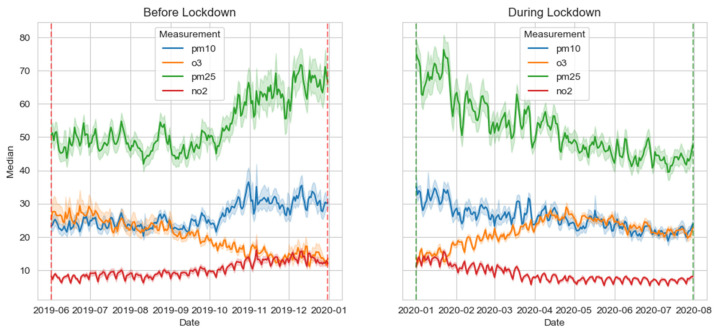
Time series of changes in pollutant levels before and during the COVID-19 lockdown period.

**Figure 5 ijerph-21-01171-f005:**
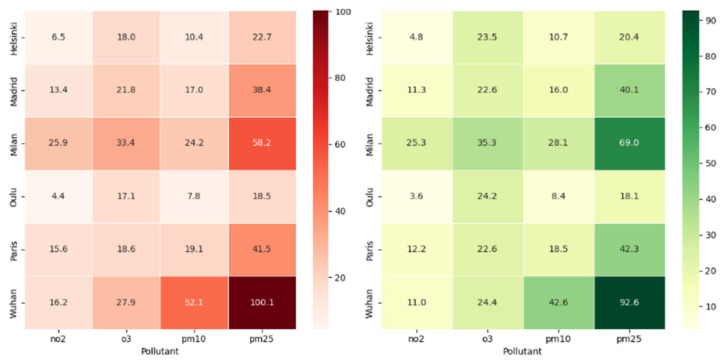
Kruskal Wallis comparison of air pollutant concentrations BL (before lockdown) and DL (during lockdown).

**Table 1 ijerph-21-01171-t001:** Interpretation of Pearson’s correlation coefficient for linear relationships.

Pearson’s Correlation Coefficient	Interpretation
+1	Perfect positive linear relationship
−1	Perfect negative linear relationship
0	Absence of a linear relationship

## Data Availability

Data for the statistical analysis are available and can be provided upon request.

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
