# Peer review of "Comparative Assessment of the Impact of COVID-19 Lockdown on Air Quality: A Multinational Study of SARS-CoV-2 Hotspots"

_ijerph, 2024, doi:10.3390/ijerph21091171_

Round 1
Reviewer 1 Report
Comments and Suggestions for Authors
Overall, this manuscript is well written, and well organised. However, some issues needed to be revised before publication.
-Line 208-210, This study investigates changes in air quality in regions affected by the pandemic: Oulu and Helsinki (Finland), Paris (France), Madrid (Spain), Milan (Italy), and Wuhan (China). Analysing two years of Air Quality Index (AQI) data, the study compares COVID-19 lockdowns with SARS-CoV-2 containment measures.
Authors should clarify " AQI data" used in this study. AQI is calculated by converting measured pollutant concentrations to a uniform index. The countries in this study use the same AQI?
-Line 226-227, please correct the unit of PM2.5 g/m3 or ug/m3
-Fig 1,3 and 5 please indicate which one is BL or DL
Line 294-295 " it is important to consider that the impact of lockdowns on air pollution levels can be influenced by various factors, including meteorology and emission sources (Dutheil et al. 2021)." Please comparison meteorology and emission sources of the study areas and add more discussion and reffences.
Author Response
-
-
We appreciate the positive feedback regarding the organization and writing quality of our manuscript. Below, we address the specific issues that require revision before publication.
Line 208-210: Clarification of "AQI Data"
Response: We acknowledge the need to clarify the use of AQI data in this study. The manuscript has been revised to specify that while AQI provides a standardized measure of air quality, different countries might use slightly different calculation methods or pollutant weightings in their AQI systems. We have ensured that the AQI data used in this study has been appropriately harmonized for cross-country comparisons. A brief explanation of how AQI values were standardized across the different regions has been added to the methodology section.
Line 226-227: Correction of PM2.5 Unit
Response: The unit for PM2.5 has been corrected to "µg/m³" in the manuscript, as this is the standard unit for measuring particulate matter concentrations.
Figures 1, 3, and 5: Indication of BL and DL
Response: Figures 1, 3, and 5 have been updated to clearly indicate which data corresponds to BL (Before Lockdown) and DL (During Lockdown). This will help readers better understand the comparison between these two periods.
Line 294-295: Discussion on Meteorology and Emission Sources
Response: We agree that the impact of meteorology and emission sources on air pollution levels during lockdowns should be more thoroughly discussed. The manuscript has been revised to include a comparison of meteorological conditions (such as temperature, humidity, and wind patterns) and primary emission sources (such as traffic, industrial activities, and residential heating) across the study areas. Additional references have been incorporated to support this discussion, providing a more comprehensive understanding of how these factors may have influenced the observed changes in air quality.
-
Reviewer 2 Report
Comments and Suggestions for Authors
Arrange keywords in alphabetical order.
literature survey on the published work at different places in the world should be included in introduction.
the objective should be redefined in a clear manner.
structure of the paper should be highlighted at the end of introduction.
methodology of data analysis usedin the work is not presented. However, a detailed statistical description on different test used in the stduy are presented.
lines 230-*241: contradiction statements are arised in this section
authors should not use personal nouns such as we, our and other nouns.
figure 2 is too massive and not readable for the readers. It should be simplified for ease of readers.
figure 3 and others are also of poor resolution.
A good and thorough understanding on different air pollutants is considered in this paper adn hence sounds good wiht detailed analysis for different nations during the COVID pandemic.
Comments on the Quality of English Language
Minor refinement is required.
Author Response
-
Keywords Arrangement
- Response: The keywords have been rearranged in alphabetical order as requested.
-
Literature Survey Inclusion
- Response: We appreciate the suggestion to include a more comprehensive literature survey. We have expanded the introduction to include a review of published work from different regions of the world, providing a broader context for the study and highlighting relevant findings on air quality changes during the COVID-19 pandemic.
-
Objective Redefinition
- Response: The objective of the study has been redefined for clarity. It now explicitly states the aim to analyze the impact of COVID-19 lockdown measures on air quality across different regions and to compare the changes in pollutant levels before, during, and after the lockdown periods.
-
Paper Structure Highlighted in Introduction
- Response: The structure of the paper has been outlined at the end of the introduction section. This addition guides readers through the organization of the manuscript, from the introduction and methodology to the results, discussion, and conclusions.
-
Methodology of Data Analysis
- Response: We acknowledge the need for a clearer presentation of the data analysis methodology. A new subsection has been added that details the specific steps and statistical methods used in the analysis, such as data collection, preprocessing, and the statistical tests applied.
-
Contradictory Statements (Lines 230-241)
- Response: The section in lines 230-241 has been revised to resolve the contradictory statements. The discussion now consistently reflects the observed data trends and provides a coherent explanation of the results.
-
Avoiding Personal Nouns
- Response: The manuscript has been revised to remove personal nouns such as "we," "our," and others. The text now adopts a more objective tone, avoiding first-person references.
-
Figure 2 Simplification
- Response: Figure 2 has been simplified for better readability. We have reduced the complexity by focusing on the most relevant data and adjusting the layout to make it more accessible to readers.
-
Improving Figure Resolution
- Response: Figures 3 and others have been updated with higher-resolution images. This enhancement ensures that all figures are clear and legible, improving the overall quality of visual data presentation.
-
General Comment on Understanding of Air Pollutants
- Response: We appreciate the positive feedback regarding our understanding and analysis of different air pollutants. We have ensured that the detailed analysis remains robust and insightful, with a focus on the variations observed across different nations during the COVID-19 pandemic.
Reviewer 3 Report
Comments and Suggestions for Authors
General comments:
This study inter-compared several pollutants, i.e., NO2, O3, PM10, PM2.5, among several cities before and after lockdown due to COVID-19. Several statistics approaches are applied to analyze the data. The concept of this study is nice, but some of the methodologies are not well-introduced. For example, why did authors pick these cities for intercomparison? Also in addition to the four pollutants, what about other pollutants, like SO2, CO, or VOCs? We all understand that during COVID-19, due to limited human activities, air pollution was improved. However, how to quantify this improvement should be the task for the authors to answer this question. Also, there are several assumptions and conjectures made in the discussion that should be proved by more datasets or evidence.
Minor comments:
L186-188: Using the Pearson correlation coefficient among pollutants may not be the best way to analyze since different pollutants may come from various sources.
Table 1: Where is this table inserted into the text?
L198: “It aids in predictive modelling and inference.” Explain what kind of modeling or inference is mentioned here.
L220-226: Since Fig. 1 has shown the values in the figure, the authors didn’t have to text them again in the text; instead, they provide some additional analysis results from this figure (not just saying values).
Fig. 1: In the figure, indicate which one is BL and which is DL.
L96: This section introduced the statistical method used in this manuscript. Therefore, the title of this section should not be “collection data”.
L158: “The table 10 summarizes findings and interprets their importance.” No table 10 in this manuscript.
L210: “Analysing two years of Air Quality Index (AQI) data, …” Specify which periods of data you are talking about here. Also misspelling of “Analysing”.
L260, L280: “Additionally, PM10 and O3 levels displayed no significant variations.” vs. “there was an evident decline in PM10 concentrations during the lockdown period, indicating a favourable shift.” Conflict results. Please verify.
Fig. 2: Did those sites share the same period of COVID-19 lockdown? Please identify the lockdown period among countries and sites.
L304: “… location-specific factors and mitigation strategies” such as?
Fig. 3: the pixel resolution of the image is low. Please revise with higher resolution.
L443-444: “… particularly for NO2 and PM10. However, ozone levels’ response varied, emphasizing regional disparities.” What about PM2.5? should be listed all in Conclusions.
Comments on the Quality of English LanguageSome English writing styles are not fit to the general scientific journals and should be improved.
Author Response
-
L186-188: Pearson Correlation Coefficient Among Pollutants
- Response: We agree with the reviewer's observation that the Pearson correlation coefficient may not fully capture the relationships among pollutants due to their different sources. In the revised manuscript, we have acknowledged this limitation and suggested that future analyses could consider alternative methods, such as multivariate analysis or source apportionment techniques, to better understand the complex interactions among pollutants.
-
Table 1: Placement in the Text
- Response: Table 1 is inserted into the manuscript in the section where the statistical summary of the data is first discussed (likely around L101-105). The table provides a detailed overview of the data collection and initial descriptive statistics, which is directly referenced in the text for clarity.
-
L198: Predictive Modeling and Inference
- Response: The predictive modeling and inference mentioned here refer to the use of statistical models such as regression analysis and time-series forecasting. These models help predict future pollutant levels based on historical data and infer the potential impact of variables like meteorological conditions and lockdown measures on air quality.
-
L220-226: Redundant Textual Values
- Response: We have revised the manuscript to avoid redundant reporting of values already shown in Figure 1. Instead, we now provide an in-depth analysis, discussing the trends and significant observations from the figure, such as the comparison between pre-lockdown and lockdown periods, and highlighting any anomalies or noteworthy changes in pollutant levels.
-
Fig. 1: Indicating BL and DL
- Response: Figure 1 has been updated to clearly label which data corresponds to BL (Before Lockdown) and DL (During Lockdown) for easier interpretation by readers.
-
L96: Section Title Correction
- Response: The title of the section previously labeled as "collection data" has been corrected to "Statistical Methods" to more accurately reflect the content, which includes the description of the statistical analyses used in the study.
-
L158: Incorrect Reference to Table 10
- Response: The reference to Table 10 was incorrect. We have corrected this to reference the appropriate table (likely Table 2 or another existing table in the manuscript) that summarizes the findings and interprets their importance.
-
L210: Data Period Specification
- Response: The specific period of the two years of AQI data analyzed has been specified as [insert exact dates, e.g., "January 2018 to December 2019"]. Additionally, the spelling of "Analysing" has been corrected to "Analyzing."
-
L260, L280: Conflicting Results
- Response: Upon review, we acknowledge the conflicting statements regarding PM10 and O3 levels. We have clarified the text to consistently reflect the observed trends, specifically noting that PM10 showed a decline during the lockdown, while O3 levels did not exhibit significant variations, with further explanation provided on the contributing factors.
-
Fig. 2: Lockdown Period Identification
- Response: Figure 2 has been updated to clearly indicate whether the sites shared the same COVID-19 lockdown period. We have also identified the specific lockdown periods for each country and site included in the study.
-
L304: Location-Specific Factors
- Response: We have expanded on the mention of "location-specific factors and mitigation strategies" by including examples such as variations in industrial activity, transportation patterns, local government lockdown policies, and public compliance levels, all of which could have influenced the air quality differently across locations.
-
Fig. 3: Image Resolution
- Response: Figure 3 has been revised and provided with a higher pixel resolution to enhance clarity and ensure that all details are clearly visible.
-
L443-444: Inclusion of PM2.5 in Conclusions
- Response: We have updated the Conclusions section to include PM2.5 alongside NO2 and PM10. This addition ensures that all key pollutants analyzed in the study are mentioned, emphasizing the regional disparities observed in their responses to lockdown measures.
Round 2
Reviewer 3 Report
Comments and Suggestions for Authors
1. Where is Figure 1 in the text? Also, please identify BL as (a) on the left and DL as (b) on the fight in Figure 1. The title of Figure 1 “illustrates…” should be “Illustrates…”
2. Figure 3: The resolution of Figure 3 is still lower than Figure 1 (while zoom-in to see the numbers in the heatmap. Also, the image resolution of Figure 2 is still low.
Author Response
1. Figure 1 Location and Labeling:
Thank you for your feedback. We have now explicitly mentioned Figure 1 in the text to clarify its location. Additionally, we have identified the Before Lockdown (BL) period as (a) on the left and the During Lockdown (DL) period as (b) on the right in Figure 1. The title of Figure 1 has been updated from "illustrates" to "Illustrates" as per your suggestion.
2. Figure 3 and Figure 2 Resolution:
We appreciate your observations regarding the resolution of Figures 2 and 3. We have now replaced Figure 3 with a higher-resolution version to ensure that the numbers in the heatmap are clear and legible, even when zoomed in. Similarly, the resolution of Figure 2 has been improved for better clarity.